# Dietary Supplementation with Transgenic *Camelina sativa* Oil Containing 20:5n-3 and 22:6n-3 or Fish Oil Induces Differential Changes in the Transcriptome of CD3^+^ T Lymphocytes

**DOI:** 10.3390/nu13093116

**Published:** 2021-09-05

**Authors:** Annette L. West, Elizabeth A. Miles, Lihua Han, Karen A. Lillycrop, Johnathan A. Napier, Philip C. Calder, Graham C. Burdge

**Affiliations:** 1School of Human Development and Health, Faculty of Medicine, University of Southampton, Southampton SO16 6YD, UK; A.West@soton.ac.uk (A.L.W.); e.a.miles@soton.ac.uk (E.A.M.); p.c.calder@soton.ac.uk (P.C.C.); 2Department of Plant Sciences, Rothamsted Research, Harpenden AL5 2JQ, UK; Lihua.han@rothamsted.ac.uk (L.H.); johnathan.napier@rothamsted.ac.uk (J.A.N.); 3Centre for Biological Sciences, Faculty of Environmental and Life Sciences, University of Southampton, Southampton SO17 1BJ, UK; k.a.lillycrop@soton.ac.uk; 4NIHR Southampton Biomedical Research Centre, University Hospital Southampton NHS Foundation Trust and University of Southampton, Southampton SO16 6YD, UK

**Keywords:** omega-3, RNAseq, T lymphocyte, transgenic plant, fish oil, eicosapentaenoic acid, docosahexaenoic acid

## Abstract

Eicosapentaenoic acid (20:5n-3) and docosahexaenoic acid (22:6n-3) are important for leukocyte function. This study investigated whether consuming transgenic *Camelina sativa* (t*CS*O) seed oil containing both 20:5n-3 and 22:6n-3 is as effective as fish oil (FO) for increasing the 20:5n-3 and 22:6n-3 content of leukocytes and altering mitogen-induced changes to the T cell transcriptome. Healthy adults (*n* = 31) consumed 450 mg/day of 20:5n-3 plus 22:6n-3 from either FO or t*CS*O for 8 weeks. Blood was collected before and after the intervention. 20:5n-3 and 22:6n-3 incorporation from t*CS*O into immune cell total lipids was comparable to FO. The relative expression of the transcriptomes of mitogen-stimulated versus unstimulated T lymphocytes in a subgroup of 16 women/test oil showed 4390 transcripts were differentially expressed at Baseline (59% up-regulated), 4769 (57% up-regulated) after FO and 3443 (38% up-regulated) after t*CS*O supplementation. The 20 most altered transcripts after supplementation differed between test oils. The most altered pathways were associated with cell proliferation and immune function. In conclusion, 20:5n-3 and 22:6n-3 incorporation into immune cells from t*CS*O was comparable to FO and can modify mitogen-induced changes in the T cell transcriptome, contingent on the lipid matrix of the oil.

## 1. Introduction

The omega-3 polyunsaturated fatty acids (PUFAs) eicosapentaenoic acid (20:5n-3) and docosahexaenoic acid (22:6n-3) are important components of cell membranes that are required for normal tissue function and development. The proportions of these PUFAs in the lipid bilayer can influence cell function by conferring fluidity on the lipid bilayer and thus influence the activities of integral membrane proteins. For example, reducing the amount of 22:6n-3 in retinal cell membranes can impair rhodopsin activation [1] and visual function [2]. 20:5n-3 and 22:6n-3 can also modulate gene transcription via the activities of ligand-activated transcription factors [3], namely peroxisome proliferator-activated receptors, as well as by inducing changes in the epigenome [4].

Oily fish are the primary dietary source of 20:5n-3 and 22:6n-3, although meat and dairy foods may also contribute to the total intake of these PUFAs [5]. Dietary supplementation with fish oil (FO) can modify the immune response [6]. 20:5n-3 and 22:6n-3 are incorporated into peripheral blood mononuclear cells (PBMCs) in a dose-dependent manner [7] and can reach an equilibrium after 4 weeks of dietary supplementation, although immunomodulatory effects have been observed within 1 week of starting the intervention [8]. Dietary supplementation with FO can induce a modest reduction in the symptoms of inflammatory diseases such as rheumatoid arthritis [9]. Such anti-inflammatory effects involve the partial displacement of 20:4n-6 by 20:5n-3 from cell membranes and increased synthesis of less biologically active eicosanoids derived from 20:5n-3, namely 3-series prostaglandins (PG) and 5-series leukotrienes [10,11,12], as well as reduction in the production of more pro-inflammatory PGE_2_ and 4-series leukotrienes from mononuclear cells [13,14,15] and decreased secretion of leukotriene B_4_ by neutrophils and monocytes [16,17,18,19]. In addition, the amelioration of an inflammatory response by 20:5n-3 and 22:6n-3 may involve increased synthesis of anti-inflammatory and inflammation-resolving mediators, namely resolvins, protectins and maresins [20].

General consumption of 20:5n-3 and 22:6n-3 in UK adults is 50% of the 450 mg (20:5n-3 plus 22:6n-3) per day dose recommended by the UK Government [5,21,22]. Failure to achieve recommended intakes reflects food choices that exclude oily fish on the grounds of cost, unpalatability or personal preference, and concerns about contamination with environmental pollutants and the sustainability of marine sources [5,21]. 20:5n-3 and 22:6n-3 intakes by individuals who exclude all animal-derived foods may be close to zero [23]. Fish oil (FO) is a supplemental source of 20:5n-3 and 22:6n-3. A transgenic strain of *Camelina sativa* has been developed that produces a seed oil (t*CS*O) containing similar amounts of both 20:5n-5 and 22:6n-3 as found in FO [24], and it is as effective as FO as a source of these fatty acids in the human diet [25,26]. However, it is not known whether this seed oil can induce changes in leukocytes that are comparable to those associated with increased consumption of FO or oily fish.

To address this, we compared the effect of dietary supplementation with t*CS*O or FO on the incorporation of 20:5n-3 and 22:6n-3 into PBMCs and on mitogen-induced changes in the T cell transcriptome.

## 2. Methods

### 2.1. Preparation of Seed Oil from Transgenic C. sativa

t*CS*O was prepared as described [25]. Briefly, homozygous T3 generation transgenic *C. sativa* plants producing a seed oil containing 20:5n-3 and 22:6n-3 [24] were grown in a biological containment glasshouse with 16 h light/8 h dark cycles (illumination of 400 μmol/m^2^ per second) and 23 °C (day) to 18 °C (night) cycles at 50–60% relative humidity [25]. Seeds were harvested and threshed, and the oil was then extracted and processed for human consumption by POS Bio-Sciences [25].

### 2.2. Dietary Supplementation

The details of the dietary supplementation study have been reported previously [25]. Briefly, the 31 healthy men and women who enrolled in the study were between 18 and 75 years of age, with a BMI between 18.5 and 30.0 kg/m^2^, were normotensive and had non-fasting total cholesterol and glucose concentrations within the normal ranges (Table 1). Volunteers who did not meet these criteria or habitually consumed any dietary oil supplements or more than one oily fish meal per week, who were unable to provide written informed consent, who were pregnant or intending to become pregnant, who were receiving treatment for a clinician-diagnosed chronic illness or who had any food allergy were excluded. Because the acute incorporation of 20:5n-3 and 22:6n-3 into blood lipids did not differ between sexes [25], the participants were studied as a single group of mixed ages and sexes (Table 1). In order to match the groups assigned to each test oil, a subgroup of 16 women, from which the highest T cell RNA yield was produced, were selected for RNAseq analysis (Table 1). Samples collected from men were not included in these analyses because of insufficient numbers of male participants to match the groups.

The trial had a single blinded crossover design, because it was not possible to conceal the odour of the FO from the participants [25]. Those who took part were randomised using a random number generator (www.random.org, accessed on January 2019) to consume either the t*CS*O (2.4 mL/day) or a commercial blended FO (Simply Timeless^®^, Omega-3 Fish Oil plus Cod Liver Oil; Seven Seas, Weybridge, Surrey, UK; 1.6 mL/day) in random order for 56 days followed by 42 days of washout, during which participants consumed their habitual diet and then consumed the other test oil for 56 days. However, because of the risk of carry over between the phases of supplementation, only data for samples collected during the first period of supplementation are reported here. The test oils were matched to provide approximately 450 mg of 20:5n-3 plus 22:6n-3 per day based on the amount recommended by the UK Government [22]. Daily and cumulative intakes of fatty acids from the test oils are shown in Table 2. Participants dispensed the appropriate volume of test oil using an oral dosing syringe and consumed this with food. Compliance was assessed by the change in weight of the bottles containing the test oils compared to the expected weight change. After fasting for 12 h, venous blood samples (40 mL) were collected into tubes containing lithium heparin anticoagulant at the start and end of each period of supplementation and separated into plasma and cell fractions [25]. PBMCs were isolated by using Histopaque (Sigma-Aldrich, Poole, Dorset, UK) density gradient centrifugation [27]. Plasma and PBMCs were stored at −80 °C before the analysis of fatty acid composition. Any participants who exhibited symptoms of infection were asked to postpone blood sampling until they were asymptomatic and then for a further 14 days.

### 2.3. T Cell Isolation and Culture

For RNAseq analysis, CD3^+^ T cells were purified from PMBCs by negative selection using the EasySep™ Human T Cell Isolation Kit (StemCell Technologies, Cambridge, Cambridgeshire, UK) according to the manufacturer’s instructions. The proportion of CD3^+^ cells in the T cell fraction was determined by flow cytometry and found to be between 80.4% and 87.7%.

Cells were cultured for 48 h in RPMI-1640 medium (Sigma-Aldrich, Poole, Dorset, UK) containing L-glutamine (2 mM), penicillin (100 units/mL) and streptomycin/(100 µg/mL), as well as 10% (*v*/*v*) autologous plasma with or without 10 µg/mL concanavalin A (Sigma-Aldrich, Poole, Dorset, UK). At 48 h, cells were collected by centrifugation, washed with PBS and the pellets were either frozen directly or resuspended in RNAlater (ThermoFisher Scientific, Oxford, Oxfordshire, UK) and then stored at −80 °C.

### 2.4. Analysis of Fatty Acid Composition by Gas Chromatography

Fatty acid analysis of total PBMC lipids was carried out as described [26]. Frozen PBMCs were thawed, and total cell lipids were then extracted with chloroform:methanol (2:1, *v*/*v*) containing butylated hydroxytoluene (50 mg/mL) [28]. Fatty acid methyl esters (FAMEs) were synthesised by reaction with methanol containing (2% *v*/*v*) concentrated sulphuric acid at 50 °C for 2 h. The reaction was cooled to room temperature and neutralised with a solution of KHCO_3_ (0.25 M) and K_2_CO_3_ (0.5 M). FAMEs were collected by repeated extraction of the reaction mixture with hexane [29] and resolved on a BPX-70 fused silica capillary column (30m × 0.25mm × 25 μm) using an Agilent 6890 gas chromatograph, (Agilent, Stockport, Cheshire, UK) equipped with flame ionisation detection [25]. Fatty acids were identified by retention times relative to standards (37 FAMES, Sigma-Aldrich, Poole, Dorset, UK). PBMC fatty acids are expressed as a proportion of total cell fatty acids.

### 2.5. Analysis of the T Cell Transcriptome by RNA Next-Generation Sequencing

Total RNA was isolated using mirVana RNA extraction kits (Invitrogen, Oxford, Oxfordshire, UK) and then frozen at −80 °C, before being transported on dry ice for RNAseq analysis. The purity of RNA was assessed using a 2100 Bioanalyser (Agilent). All samples had RIN scores greater than 8.5. RNA was quantified using the Qubit RNA HS Assay Kit (ThermoFisher Scientific).

RNA sequencing was carried out by Omega Bioservices (Norcross, GA, USA). Libraries were prepared using the TruSeq^®^ Stranded mRNA Library Preparation kit (Illumina, San Diego, CA, USA) and quantified using the Promega QuantiFluor dsDNA System on a Quantus Fluorometer (Promega, Madison, WI, USA). The size and purity of the libraries were analysed using the High Sensitivity D1000 Screen Tape on an Agilent 2200 TapeStation instrument. The libraries were normalised, pooled and clustered, and pair read sequencing was performed for 150 cycles on an Illumina HiSeqX10 instrument (150 read cycles).

The quality control, normalisation and initial data analysis were carried out by Omega Bioservices using the Illumina Basespace package (Version 2.0.1; www.illumina.com/products/by-type/informatics-products/basespace-sequence-hub/apps/rna-seq-alignment.html). Sequencing reads were aligned to the *Homo sapiens* reference genome UCSC hg19 using the STAR RNA sequence aligner [30]. Reference genes and transcripts were quantified using the Salmon software package [31]. The Strelka variant caller was used to identifies single nucleotide variants and small indels [32]. Gene counts, gene FPKMs, principal component analysis and differential expression were produced using DESeq2 [33]. Transcripts that differed between stimulated and unstimulated cells by at least 1.2-fold with an adjusted *p* < 0.05 were considered to be differentially expressed. Pathway analysis was carried out in house using the Ingenuity Pathway Analysis package. Pathways that differed between unstimulated and stimulated cells by at least 1.2-fold with *p* < 0.05 and enriched in ≥10 differentially expressed transcripts were considered to be significantly altered. RNA sequencing data were validated by RT-qPCR using the SyberGreen method as described [34] using the primers listed in Appendix A. Five genes were chosen at random that differed significantly between unstimulated and stimulated T lymphocytes at Baseline and represented the range of mitogen-induced changes detected by the arrays.

### 2.6. Statistical Analysis

Fatty acid composition data are presented as mean ± SEM, unless stated otherwise, and analysed by two-way ANOVA with the type of oil as a fixed factor and time as a repeated measure with Tukey’s post hoc correction using IBM SPSS Statistics for Windows (Version 26.0, IBM, Armonk, NY, USA).

## 3. Results

### 3.1. Participant Characteristics

In total, 32 men and women were enrolled, of which 31 completed the study (Table 1). One participant withdrew from the study because they could not tolerate the taste of the FO [25] and was consequently excluded from all analyses. There were no significant differences between groups in the characteristics of the participants.

### 3.2. Provision of Fatty Acids from Test Oils

The provision of saturated fatty acids (SFAs) per day by t*CS*O was similar to that from FO (Table 2). FO provided 1.5-fold more 16:0 than t*CS*O, while t*CS*O provided 3-fold more 18:0 than FO. The amount of total monounsaturated fatty acids (MUFAs) provided by FO was 1.5-fold greater (due primarily to greater amounts of 16:1n-7 and 18:1n-9) than that provided by t*CS*O (Table 2). The total amount of n-6 PUFAs provided by t*CS*O was 13-fold greater than that provided by FO due, mainly due to the 20-fold higher 18:2n-6 content of the t*CS*O than FO (Table 2). The total amount of n-3 PUFAs provided by t*CS*O was 1.7-fold greater than that provided by FO, due mainly to the higher 18:3n-3 (12-fold) and 22:5n-3 (5-fold) contents of the t*CS*O, while the 20:5n-3 and 22:6n-3 contents were similar between the test oils (Table 2).

### 3.3. Effect of Dietary Supplementation on the Fatty Acid Composition of PBMCs

Dietary supplementation induced modest, statistically significant reductions in the proportions of 18:1n-7 (t*CS*O −7.1%; FO −8%), 18:2n-6 (t*CS*O −9%; FO −3%) and 18:3n-6 (t*CS*O −20%; FO −40%) in PBMC total lipids (Table 3), although there was no significant effect of the type of test oil. Dietary supplementation induced significant increases in the proportions of 20:5n-3 (t*CS*O 13%; FO 13%) and 22:6n-3 (t*CS*O 5%; FO 16%) in PBMC total lipids (Table 3). There was no significant effect of the type of test oil consumed on the proportions of 20:5n-3 and 22:6n-3. However, there was a significant time × test oil effect on the proportion of 22:5n-3, such that there was a significant increase (29%) following t*CS*O supplementation with 22:5n-3, but no detectable change after consuming FO (Table 3). There was no significant effect of dietary supplementation or the type of test oil on the proportion of any of the other fatty acids measured in PBMCs (Table 3). There were no significant differences before or after dietary supplementation with either test oil on the fatty acid composition of PBMCs between the subgroups, from which T cells were analysed by RNAseq and the corresponding whole groups (Appendix A).

### 3.4. The Effect of Dietary Supplementation on the Number of Differentially Expressed Transcripts in Activated Compared to Unstimulated CD3^+^ T Cells

The findings of the RNAseq analysis were validated by RT-qPCR. The five genes tested showed a significant difference between stimulated and unstimulated cells in the same direction by both techniques (Appendix A).

Mitogen stimulation of T cells collected at Baseline induced differential expression of 4,310 transcripts, of which 2517 (59.4%) were up-regulated compared to unstimulated cells (Figure 1A,B). Mitogen stimulation after FO supplementation induced differential expression compared to unstimulated cells of 4769 transcripts, of which 2723 (57.1%) were up-regulated (Figure 1A,B). Mitogen stimulation after t*CS*O supplementation induced differential expression compared to unstimulated cells of 4652 transcripts, among which 2514 (54.0%) were up-regulated (Figure 1A,B). In total, 2758 transcripts were expressed differentially at Baseline and after supplementation with either test oil, of which 65.8% were up-regulated.

Mitogen stimulation at Baseline induced up-regulation of 303 (12.0%) transcripts that did not overlap with transcripts that were differentially expressed after supplementation with either test oil (Figure 1A). Mitogen stimulation after FO supplementation induced up-regulation of 477 (17.5%) transcripts that did not overlap with those up-regulated at Baseline or after t*CS*O supplementation. Mitogen stimulation after t*CS*O supplementation induced up-regulation of 320 (12.7%) transcripts that did not overlap with those up-regulated at Baseline or after FO supplementation (Figure 1A). Overall, 2,014 transcripts that were up-regulated at Baseline were also up-regulated after FO supplementation, while 1,989 transcripts that were up-regulated at Baseline were also up-regulated after t*CS*O supplementation. Further, 2,021 transcripts were up-regulated by supplementation with both test oils, of which 1,986 (98.2%) were also up-regulated at Baseline (Figure 1A).

Mitogen stimulation at Baseline induced down-regulation of 396 (22.1%) transcripts that did not overlap with transcripts that were down-regulated after supplementation with either test oil (Figure 1B). Mitogen stimulation after FO supplementation induced down-regulation of 616 (30.1%) transcripts that were not down-regulated at Baseline or after t*CS*O supplementation. Mitogen stimulation after t*CS*O supplementation induced down-regulation of 631 (29.5%) transcripts that were not down-regulated at Baseline or after FO supplementation (Figure 1B). In total, 1,241 (61%) transcripts that were down-regulated after FO supplementation were also down-regulated by supplementation with t*CS*O (Figure 1B).

In this study, 45 (1.8%) of the transcripts up-regulated in mitogen-stimulated compared to mitogen-unstimulated T cells at Baseline were down-regulated after supplementation with FO. Additionally, 25 (1%) of the transcripts up-regulated in mitogen-stimulated compared to unstimulated T cells at Baseline were down-regulated after supplementation t*CS*O (Figure 2A). Furthermore, 34 (1.8%) transcripts that were down-regulated at Baseline were up-regulated after FO supplementation; 22 (1.2%) transcripts that were down-regulated at Baseline were down-regulated after t*CS*O supplementation. Six transcripts that were up-regulated by mitogen stimulation at Baseline were down-regulated after supplementation with either FO or t*CS*O, namely *INSM2*, *C1orf105*, *DKFZp779M0652*, *CLDN5*, *PPIAL4F* and *THRB* (Figure 2A). Three transcripts that were down-regulated at Baseline were up-regulated after supplementation with either FO or t*CS*O, namely *GOLGA8S*, *TRIM39-RPP21* and *EPHX3* (Figure 2B).

### 3.5. The Effect of Dietary Supplementation on the Most Differentially Expressed Transcripts in Activated Compared to Unstimulated CD3^+^ T Cells

Four transcripts were among the 20 most up-regulated transcripts at Baseline and after supplementation with either FO or t*CS*O, namely interleukin (IL)-17F, IL-5, Kinesin Family Member 1A (*KIF1A*) and Thymidylate Synthetase (*TYMS*) (Table 4). In addition, Baculoviral Inhibitor of Apoptosis Repeat-Containing 5 (*BIRC5*) and Cystathionine β-Synthase-like protein (*CBSL*) were among the 20 most differentially up-regulated transcripts after supplementation with FO or t*CS*O, but not in cells collected at Baseline (Table 4). Gap junction protein-Beta-2 (*GJB2*) and Minichromosome Maintenance 10 Replication Initiation Factor (*MCM10*) were among the most up-regulated transcripts at Baseline and after supplementation with t*CS*O, but not with FO (Table 4). Zinc-Finger BED-Type Containing-2 (*ZBED2*), IL-22 and spindle pole body component 25 (*SPC25*) were among the 20 most up-regulated transcripts at Baseline and after supplementation with FO, but not with t*CS*O (Table 4).

### 3.6. Effect of Dietary Supplementation on Differentially Expressed Pathways in Activated CD3^+^ T Cells

The number of significantly enriched pathways in mitogen-stimulated compared to unstimulated cells (≥10 differentially expressed transcripts; ≥1.2-fold difference from unstimulated cells, α < 0.05) was greater following t*CS*O supplementation (617 pathways) than after FO supplementation (601 pathways) or at Baseline (517 pathways).

The 25 most altered pathways were either associated with immune cell function or cell proliferation (Table 5, Figure 3). Of these, 14 pathways were differentially expressed in stimulated compared to unstimulated cells at Baseline and after supplementation with either test oil (Table 5), and 3 pathways were differentially expressed at Baseline and after FO supplementation, while 3 different pathways were differentially expressed at Baseline and after t*CS*O supplementation. Additionally, three pathways were differentially expressed at Baseline alone, three others were differentially expressed only after FO supplementation and five pathways were differentially expressed after t*CS*O, but not after FO supplementation or at Baseline (Table 5).

Dietary supplementation with either FO or t*CS*O increased the proportions of up-regulated transcripts in 18/25 of the most altered pathways identified at Baseline (Figure 3). The proportion of up-regulated transcripts in the ‘Role of cytokines in mediating communication between immune cells’ pathway was similar irrespective of dietary supplementation (Figure 3A). The proportion of up-regulated transcripts in breast cancer regulation by the stathmin-1 pathway was similar at Baseline to after supplementation with t*CS*O, but approximately 3-fold greater after FO supplementation (Figure 3B).

## 4. Discussion

The findings show that incorporation of 20:5n-3 and 22:6n-3 into PBMC total lipids from t*CS*O was similar to that from FO when these fatty acids are consumed according to the daily recommendation made by the UK Government to maintain general health [22]. Despite equivalence in the incorporation of 20:5n-3 and 22:6n-3 into PBMCs, there were differences between test oils in the mitogen-induced changes to the expression of the CD3^+^ T cell transcriptome.

We have shown that dietary supplementation with either t*CS*O or FO induced comparable increments of 20:5n-3 and 22:6n-3 concentrations in individual plasma lipid classes in samples from the present study [25]. Since leukocytes can assimilate fatty acids from both lipoprotein [35] and non-esterified fatty acid pools [36], these findings show an overall enrichment of 20:5n-3 and 22:6n-3 in the lipid environment to which T cells were exposed in vivo. Consequently, 20:5n-3 and 22:6n-3 were also enriched in PBMC lipids, independent of the type of oil supplement. The proportions of 22:5n-3, 16:1n-7, 18:2n-6, 18:3n-6 and 20:2n-6 were also increased in quiescent PBMCs after supplementation with the test oils, although they were not increased in plasma lipids [25]. Leukocytes can preferentially accumulate PUFAs from their environment [37]; however, the composition of leukocyte cell membranes reflects primarily the specificity of phospholipid biosynthesis [38]. The proportions of 22:5n-3, 16:1n-7, 18:2n-6, 18:3n-6 and 20:2n-6 increase in activated leukocytes, particularly T cells [39,40,41], as a result of changes in the specificity of phospholipid biosynthesis including increased phospholipid acyl-remodelling processes [37,38]. One possible explanation for enrichment of 22:5n-3, 16:1n-7, 18:2n-6, 18:3n-6 and 20:2n-6 in PBMCs in the present study is that it reflects induction of membrane changes by partial activation of the cells due by cell processing in vitro.

There is considerable evidence that consumption of oils containing 20:5n-3 and 22:6n-3 can suppress the immune response [20] and modify the mRNA expression of individual genes that are involved in immune cell function [42,43,44]. However, there is limited information about the effect of dietary supplementation with oils containing 20:5n-3 and 22:6n-3 on activation-associated changes in the T cell transcriptome. Previous analyses of the effect of dietary supplementation with FO or seed oils showed differential changes in the expression of the transcriptome of PBMCs. FO supplementation has been shown to increase the mRNA expression of genes involved in cell cycling in quiescent PBMCs compared to a sunflower oil supplement [45]. Others have shown that consumption of 1.8 g/day of 20:5n-3 plus 22:6n-3 for 26 weeks in elderly men and women induced the differential expression of 1,040 genes in quiescent PBMCs, primarily down-regulation of transcripts involved in inflammatory and atherogenic pathways, compared to 298 differentially expressed genes following supplementation with 4 g/day of sunflower oil [46]. Moreover, Polus et al. found the differential expression of genes involved in inflammation and lipid metabolism in whole blood from obese women after supplementation with 1.8 g/day of 20:5n-3 plus 22:6n-3 compared to an undisclosed placebo [47]. None of these studies investigated the effect of dietary supplementation on changes induced in the leukocyte transcriptomes by mitogen stimulation. Therefore, it is uncertain whether the differences in gene expression in quiescent PBMCs are reflected in activated cells or if they related to leukocyte function. Furthermore, because these studies investigated the transcriptome of PBMCs, instead of single cell types, the findings are at risk of confounding due to variation in the relative proportions of each cell type both between individuals and within the same participant over time. None of the previous studies disclosed whether the data were corrected for the proportions of different types of leukocytes.

The present study showed differential effects between the test oils and compared to Baseline on mitogen-induced changes in individual genes and pathways in CD3^+^ T lymphocytes. These findings suggested that consuming these test oils altered the capacity of T cells to undergo changes in transcription that are associated with the immune response. Only 20% of the most up-regulated and 35% of the most down-regulated transcripts were among the most differentially expressed at Baseline and after supplementation with either test oil. This suggests that, despite comparable changes in PBMC lipid composition, these test oils differed in their effects on the expression of the transcriptome. Moreover, the direction of mitogen-induced change in expression was reversed for a proportion (<2%) of differentially expressed genes irrespective of the test oil supplement. Two of the most down-regulated transcripts were considered anomalies, namely CD163, which is a macrophage-specific scavenger receptor [48], and CD1d, a marker expressed by antigen-presenting cells including CD19^+^ CD1d^+^ regulatory B lymphocytes [49] that is also a target for invariant natural killer T cells [50]. One possible explanation is that there was low-level contamination of the T cell preparations with macrophages or CD1d^+^ B cells that was not detected by routine assessment of the purity of the T cell preparation by flow cytometry. If so, the decrease in the expression of these markers could be explained by reduction in the proportion of contaminating cell populations by the mitogen-induced expansion of CD3^+^ T lymphocytes.

The most altered pathways were associated with cell proliferation and immune response in stimulated compared to unstimulated cells. A previous RNAseq analysis of PBMC cultures stimulated with anti-CD3 antibodies also showed increased expression of pathways involved in immune function and cell division [51], although differences between this previous report and the present findings may be due to analysis of a mixture of cell types in the previous study. Both FO and t*CS*O increased the number of up-regulated transcripts in pathways involved in cell division [52] and immune-regulatory pathways such as interleukin-17 and interleukin-10 signalling [20,53]. This suggests that even modest 20:5n-3 and 22:6n-3 intakes might modify immune function, possibly by changing the balance between cross-talking pathways [54].

Inevitably, there are some limitations to the study design. These include the modest number of participants, although groups were well-matched, which was based on a previous investigation of the effect of FO supplementation on the fatty acid composition of PBMCs [7], as there are no previous studies of the effect of supplementation with FO on measurements of the expression of the T cell transcriptome by RNAseq on which to base a statistical power calculation. For this reason, the current findings could be regarded as exploratory. It is possible that there may have been low-level contamination of the T lymphocyte preparations that was not detected by the limited assessment of the cell composition. There were no analyses of cell function, such as cytokine secretion, because these measurements were not required to address the primary aim of the study. Nevertheless, previous studies have shown that omega-3 PUFA-induced changes in cytokine mRNA levels, including IL-17A and IL-5 in human [55] and murine [56] T cells or PBMCs, are related directly to their concentration in cell culture supernatants. Therefore, although the present study did not investigate whether changes in cytokine mRNA levels were associated with differences in their respective concentrations, it is probable that differences in gene expression would have altered cytokine secretion.

These analyses do not permit deduction of the underlying mechanism by which such differential effects between test oils were induced. Although different individuals were randomised to either test oil, these groups were composed of women who were well matched with similar compliance, and so any effect of confounding due to different participants in each arm of the trial is likely to be modest. The quantities of the test oils that were consumed were designed to deliver similar amounts of 20:5n-3 plus 22:6n-3, but they were not matched for the other fatty acids present. For example, the amounts of n-6 PUFAs, primarily 18:2n-6, were approximately 10-fold greater in t*CS*O than in FO. These differences in the fatty acid composition of the test oils were not reflected in the composition of PBMC total lipids. This suggest that any influence of difference in the amounts of fatty acids provided between test oils on cell function is unlikely to be mediated via changes in the composition of T cell membranes alone. PPARγ activity in PBMCs has been shown to be modified by dietary supplementation with FO [43,57]. Therefore, one possible explanation for the differential effects of the test oil on the expression of the T cell transcriptome is via the actions of ligand-activated fatty acid-binding transcription factors, such as members of the PPAR family. Furthermore, dietary supplementation with FO or olive oil has been shown to induce changes in DNA methylation of PBMCs that are contingent on the type of oil [58], and the treatment of Jurkat T lymphocyte leukaemia cells with physiologically relevant amounts of 22:6n-3 or 18:1n-9 induced differential methylation of individual CpG loci [59]. This suggests that induction of altered epigenetic regulation may be one additional or alternative mechanism to any direct effect on PPAR activity by which supplementation with FO or t*CS*O could modify mitogen-induced changes in the T cell transcriptome.

## 5. Conclusions

The findings of the present study show for the first time that consuming an amount of FO or t*CS*O oil, which each provided 450 mg/day of 20:5n-3 plus 22:6n-3, the amount recommended by the UK Government for maintaining general health [22], modified the activation-induced changes in the expression of the T cell transcriptome, and thus may alter the regulation of their function. These changes to the response of the transcriptome to activation differed between test oils, which implies that 20:5n-3 and 22:6n-6 were not the only mediators of altered transcription. One possible implication of these findings is that consuming the amount of 20:5n-3 plus 22:6n-3 recommended by the UK Government [22] can induce changes in the T cell transcriptome that are relevant for understanding differences between individuals in their response to infection or vaccination. Moreover, the precise effect on the transcriptome of supplements containing EPA plus DHA may be contingent on the overall fatty acid composition of the oil.

## Figures and Tables

**Figure 1 nutrients-13-03116-f001:**
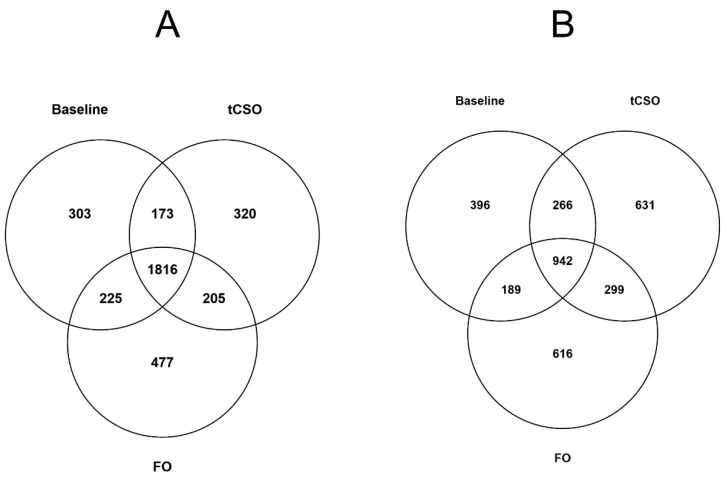
Differentially expressed transcripts between stimulated and unstimulated T cells at Baseline and after supplementation with the test oils. Values are numbers of differentially expressed transcripts (log2 fold change >1.2; *p* < 0.05) between unstimulated and stimulated cells at Baseline and after 8 weeks of dietary supplementation with either fish oil (FO) or transgenic *Camelina sativa* oil (t*CS*O). (**A**) Up-regulated transcripts and (**B**) down-regulated transcripts.

**Figure 2 nutrients-13-03116-f002:**
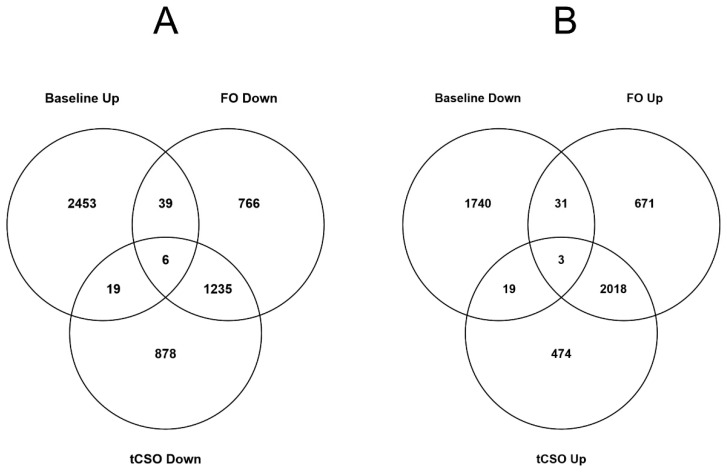
Differentially expressed transcripts between stimulated and unstimulated T cells at Baseline that showed the opposite mitogen-induced change after supplementation with the test oils. Values are numbers of differentially expressed transcripts (log2 fold change >1.2; *p* < 0.05) between unstimulated and stimulated cells at Baseline and after 8 weeks dietary supplementation with either fish oil (FO) or transgenic *Camelina sativa* oil (t*CS*O). (**A**) Up-regulated transcripts at Baseline compared to down-regulated transcripts after supplementation and (**B**) down-regulated transcripts at Baseline compared to down-regulated transcripts after supplementation.

**Figure 3 nutrients-13-03116-f003:**
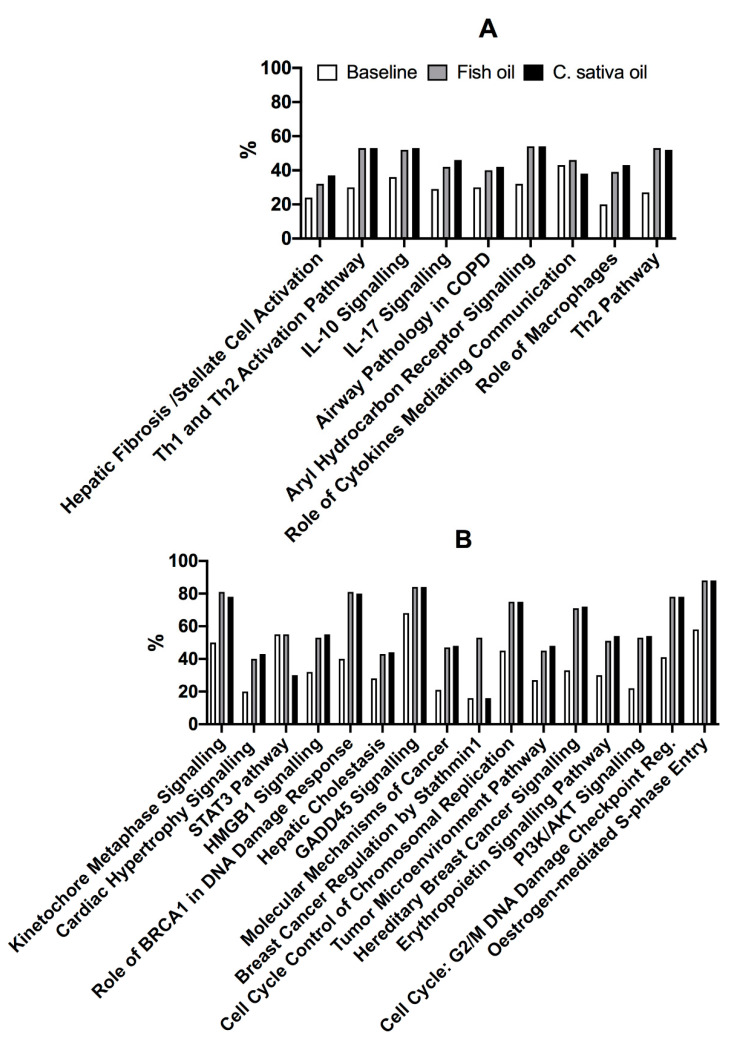
The proportions of up-regulated transcripts in the 20 most altered pathways at Baseline. Values are proportions (%) of up-regulated transcripts in stimulated compared to unstimulated cells at Baseline and after supplementation with either FO or t*CS*O in the 20 most altered pathways at Baseline grouped according to their putative roles (**A**) immune response and (**B**) cell proliferation. The names of some pathways have been shortened. The full pathway names are shown in Table 5.

**Table 1 nutrients-13-03116-t001:** Characteristics of participants. Values are mean ± SEM. There were no significant differences by Student’s unpaired *t*-test between males and females, and all females and the RNAseq subgroup at Baseline. * All participants in the RNAseq subgroup were female. ^‡^ Comparison between the FO and t*CS*O RNAseq subgroups by Student’s unpaired *t*-test.

	Whole Group	Males	Females	RNAseq Subgroup *			
				All (Baseline)	FO Group	t*CS*O Group	^‡^*t* Test
*n*	31	13	18	16	8	8	*p*
Age (years)	44.6 ± 3.4	46.6 ± 5.3	43.1 ± 4.9	44.4 ± 5.5	40.1 ± 7.9	46.1 ± 7.6	0.595
BMI (kg/m^2^)	24.5 ± 0.5	25.2 ± 0.8	23.9 ± 0.6	26.6 ± 5.5	23.2 ± 1.3	24.3 ± 0.7	0.468
Systolic bp (mm Hg)	118 ± 2	123 ± 2	114 ± 3.5	106 ± 8	117 ± 5.3	111.6 ± 5.8	0.522
Diastolic bp (mm Hg)	69 ± 2	73 ± 2	66 ± 2	65 ± 2	66.9 ± 2.8	65.3 ± 2.9	0.713
Glucose (mmol/L)	5.1 ± 0.1	4.9 ± 0.1	5.3 ± 0.2	5.4 ± 0.2	5.1 ± 0.2	5.6 ± 0.4	0.258
Cholesterol (mmol/L)	4.9 ± 0.2	4.8 ± 0.2	5.0 ± 0.3	5.2 ± 0.3	5.2 ± 0.4	4.9 ± 0.5	0.665

**Table 2 nutrients-13-03116-t002:** Fatty acid provision from test oil supplements. Values are the amounts of each fatty acid provided by t*CS*O or FO to each participant daily and the cumulative amount provided over the 56 days intervention period. MUFAs, monounsaturated fatty acids, PUFAs, polyunsaturated fatty acids, SFAs, saturated fatty acids.

Fatty Acid	Daily Provision (mg/day)		Cumulative Provision (g over 56 days)	
	FO	t*CS*O	FO	t*CS*O
14:0	100	3	5.6	0.2
16:0	232	156	13.0	8.7
18:0	45	130	2.5	7.3
20:0	3	68	0.1	3.8
24:0	2	24	0.1	1.3
Total SFAs	381	380	21.4	21.3
16:1n-7	139	5	7.8	0.3
18:1n-9	224	139	12.6	7.8
18:1n-7	63	37	3.5	2.1
20:1n-9	102	144	5.7	8.0
Total MUFAs	527	325	29.5	18.2
18:2n-6	24	468	1.4	26.2
18:3n-6	3	73	0.2	4.1
20:2n-6	4	20	0.2	1.1
20:3n-6	3	21	0.2	1.2
20:4n-6	15	67	0.9	3.8
Total n-6 PUFAs	50	650	2.8	36.4
18:3n-3	16	321	0.9	18.0
20:4n-3	96	66	5.4	3.7
20:5n-3	247	254	13.8	14.2
22:5n-3	29	155	1.6	8.7
22:6n-3	199	202	11.1	11.3
20:5n-3 + 22:6n-3	445	456	24.9	25.5
Total n-3 PUFAs	587	998	33	56
Total fatty acids	1545	2352	86.5	131.7

**Table 3 nutrients-13-03116-t003:** The effect of dietary supplementation with t*CS*O or FO in the fatty acid composition of PBMCs. Values are mean ± SEM. Data were analysed by ANOVA with time as a repeated measure and the type of oil as a fixed factor for all (*n* = 31) participants. df were 1.60. There were no single factor effects of the type of test oil. The fatty acid composition and statistical analyses of PBMCs from the subgroup of participants whose samples were used in RNAseq analyses are shown in Appendix A.

	Proportion of Total Fatty Acids				ANOVA			
	%				Time		Time × Test Oil	
	FO		t*CS*O		F	*p*	F	*p*
	Start	End	Start	End				
14:0	0.5 ± 0.1	0.6 ± 0.1	0.7 ± 0.1	0.5 ± 0.1	0.82	0.35	0.25	0.88
16:0	19.8 ± 0.9	19.9 ± 1.2	21.3 ± 1.2	18.6 ± 1.2	3.34	0.08	4.14	0.05
16:1n-7	0.3 ± 0.0	0.3 ± 0.0	0.3 ± 0.0	0.3 ± 0.0	0.04	0.85	0.42	0.52
18:0	27.7 ± 0.7	27.4 ± 0.5	27.6 ± 0.7	27.0 ± 0.8	0.83	0.37	1.21	0.73
18:1n-9	14.8 ± 0.5	15.1 ± 0.4	13.8 ± 0.5	15.3 ± 0.5	5.98	0.02	2.47	0.12
18:1n-7	1.4 ± 0.1	1.3 ± 0.0	1.3 ± 0.0	1.2 ± 0.1	6.22	0.02	0.01	0.91
18:2n-6	6.7 ± 0.3	6.1 ± 0.2	6.2 ± 0.2	6.0 ± 0.2	5.35	0.02	1.36	0.24
18:3n-6	0.5 ± 0.0	0.4 ± 0.0	0.5 ± 0.1	0.3 ± 0.0	3.895	0.04	0.42	0.52
18:3n-3	1.0 ± 0.1	1.0 ± 0.1	1.2 ± 0.1	1.1 ± 0.1	0.45	0.51	0.70	0.79
20:0	0.6 ± 0.1	0.6 ± 0.1	0.6 ± 0.1	0.5 ± 0.1	0.79	0.39	1.22	0.27
20:1n-9	1.2 ± 0.1	1.2 ± 0.1	1.3 ± 0.1	1.1 ± 0.1	1.65	0.20	1.35	0.25
20:2n-6	0.6 ± 0.1	0.6 ± 0.1	0.6 ± 0.1	0.6 ± 0.1	0.001	0.98	0.55	0.46
20:3n-6	1.8 ± 0.1	1.8 ± 0.1	1.9 ± 0.1	1.7 ± 0.1	1.1	0.31	4.88	0.07
20:4n-6	16.5 ± 0.9	16.6 ± 0.9	16.3 ± 1.1	18.4 ± 1.1	2.49	0.12	2.42	0.13
20:5n-3	0.8 ± 0.1	0.9 ± 0.1	0.8 ± 0.1	0.9 ± 0.1	4.51	0.04	0.07	0.80
22:5n-3	1.9 ± 0.1	1.9 ± 0.1	1.7 ± 0.1	2.2 ± 0.1	12.22	0.001	5.35	0.02
22:6n-3	2.1 ± 0.1	2.2 ± 0.1	1.9 ± 0.1	2.2 ± 0.1	4.20	0.04	1.13	0.30

**Table 4 nutrients-13-03116-t004:** The 20 most changed transcripts in response to stimulation with Concanavalin A in T cells collected from women either before supplementation or after supplementation with either t*CS*O or FO. Values are log2 fold difference between mitogen-stimulated and -unstimulated cells.

	Difference in mRNA Expression in Concanavalin A-Stimulated Cells and Unstimulated Cells							
	Baseline		Post t*CS*O Supplementation			Post FO Supplementation		
*n*	16		8			8		
Gene	log2 Fold Change	Adjusted *p*	Gene	log2 Fold Change	Adjusted *p*	Gene	log2 Fold Change	Adjusted *p*
	Up-regulated genes							
*CXCL9*	8.97	1.16 × 10^−32^	*KIF20A*	8.71	6.13 × 10^−73^	*CSF2*	8.50	2.48 × 10^−26^
*IL17A*	8.40	1.14 × 10^−15^	*IL17F*	8.68	9.23 × 10^−21^	*GJB2*	8.13	1.15 × 10^−12^
*IL17F*	7.84	6.67 × 10^−21^	*ZBED2*	8.60	5.08 × 10^−45^	*CAVIN3*	8.03	1.17 × 10^−17^
*GJB2*	7.80	6.24 × 10^−26^	*SPC25*	8.60	3.61 × 10^−34^	*RIBC2*	7.91	1.46 × 10^−17^
*CCL17*	7.69	5.03 × 10^−18^	*IL22*	8.57	3.52 × 10^−17^	*IL17F*	7.91	2.68 × 10^−17^
*ZBED2*	7.63	3.78 × 10^−28^	*IL17A*	8.50	2.94 × 10^−12^	*CBSL*	7.84	2.09 × 10^−11^
*CXCL10*	7.42	1.20 × 10^−18^	*XIRP1*	8.40	6.14 × 10^−26^	*MCM10*	7.76	2.56 × 10^−29^
*IL5*	7.27	1.62 × 10^−15^	*PKMYT1*	8.32	2.78 × 10^−68^	*IL5*	7.57	2.19 × 10^−07^
*HIST1H2BH*	7.20	4.41 × 10^−29^	*IL5*	8.23	5.14 × 10^−13^	*CCL1*	7.53	6.48 × 10^−16^
*IFNG*	7.19	3.46 × 10^−27^	*UHRF1*	8.17	1.65 × 10^−140^	*CDC25C*	7.49	3.57 × 10^−17^
*EBI3*	7.17	1.53 × 10^−46^	*KIF18B*	8.13	5.09 × 10^−88^	*TYMS*	7.47	3.65 × 10^−20^
*IL22*	7.15	1.04 × 10^−15^	*CCNB2*	8.11	5.83 × 10^−67^	*CDC25A*	7.40	1.26 × 10^−20^
*SPC25*	7.14	1.82 × 10^−19^	*KIF1A*	8.08	8.82 × 10^−22^	*CDC20*	7.33	3.20 × 10^−17^
*FLJ21408*	7.13	2.02 × 10^−20^	*RRM2*	8.02	2.36 × 10^−49^	*KIF1A*	7.33	1.63 × 10^−16^
*KIF1A*	7.06	8.69 × 10^−23^	*TYMS*	8.02	4.02 × 10^−78^	*FGF2*	7.29	9.95 × 10^−16^
*IL21*	7.02	4.12 × 10^−21^	*BIRC5*	7.85	5.30 × 10^−73^	*IL24*	7.29	1.20 × 10^−11^
*MCM10*	7.01	7.03 × 10^−23^	*CBSL*	7.82	1.59 × 10^−15^	*BIRC5*	7.28	5.84 × 10^−16^
*MYBL2*	7.01	1.29 × 10^−28^	*LINC01132*	7.80	4.72 × 10^−23^	*ANXA3*	7.24	3.10 × 10^−09^
*TYMS*	6.97	1.64 × 10^−25^	*GNG4*	7.76	2.32 × 10^−33^	*PBK*	7.24	2.41 × 10^−16^
*DLGAP5*	6.96	7.44 × 10^−25^	*STC2*	7.69	6.13 × 10^−20^	*SMIM11A*	7.19	0.012
	Down-regulated genes							
*TREM2*	−6.45	3.34 × 10^−13^	*RNASE1*	−8.15	5.49 × 10^−23^	*SMN1*	−26.27	5.38 × 10^−17^
*FAM198B*	−6.01	2.85 × 10^−10^	*TREM2*	−6.77	1.31 × 10^−12^	*EEF1E1_BLOC1S5*	−23.38	1.23 × 10^−13^
*GPNMB*	−5.90	2.93 × 10^−21^	*HSFX2*	−6.76	0.04	*LOC102723996*	−23.19	2.00 × 10^−13^
*RNASE1*	−5.84	2.34 × 10^−07^	*CNTNAP2*	−6.53	4.28 × 10^−10^	*RNASE1*	−7.60	8.65 × 10^−05^
*APOC2*	−5.67	3.95 × 10^−07^	*OLR1*	−6.27	5.65 × 10^−07^	*TREM2*	−6.96	2.91 × 10^−07^
*KCNJ5*	−5.63	1.03 × 10^−07^	*LINC00891*	−6.26	0.002	*GPNMB*	−6.17	8.35 × 10^−15^
*A2M*	−5.58	9.60 × 10^−17^	*F13A1*	−6.24	1.54 × 10^−11^	*LILRB5*	−5.99	6.70 × 10^−06^
*RAB42*	−5.42	4.82 × 10^−08^	*GPNMB*	−6.15	5.01 × 10^−28^	*CD163*	−5.98	1.05 × 10^−12^
*LILRB5*	−5.23	2.02 × 10^−07^	*C1QA*	−6.09	2.01 × 10^−07^	*CNTNAP2*	−5.91	4.45 × 10^−09^
*FCN1*	−5.10	2.49 × 10^−10^	*FPR3*	−6.06	6.52 × 10^−16^	*SLCO2B1*	−5.88	2.22 × 10^−08^
*DCANP1*	−5.08	1.11 × 10^−08^	*FAM198B*	−6.00	1.72 × 10^−10^	*FAM198B*	−5.83	0.0003
*STAB1*	−5.01	1.04 × 10^−13^	*GTF2IP4*	−5.85	0.02	*DCANP1*	−5.69	1.10 × 10^−05^
*ATP6V0D2*	−4.98	6.54 × 10^−07^	*STAB1*	−5.80	4.66 × 10^−21^	*MSR1*	−5.66	0.0001
*CNTNAP2*	−4.91	2.72 × 10^−15^	*HNMT*	−5.76	4.85 × 10^−08^	*CLEC10A*	−5.62	6.39 × 10^−10^
*CLEC9A*	−4.84	5.86 × 10^−07^	*A2M*	−5.74	5.33 × 10^−15^	*F13A1*	−5.57	1.77 × 10^−05^
*F13A1*	−4.79	2.04 × 10^−09^	*CLEC10A*	−5.66	5.39 × 10^−10^	*DNASE1L3*	−5.55	1.88 × 10^−06^
*APOE*	−4.72	1.20 × 10^−16^	*DCANP1*	−5.56	4.06 × 10^−06^	*THBD*	−5.53	0.001
*HS3ST2*	−4.72	0.0002	*LILRB5*	−5.37	1.51 × 10^−07^	*FABP4*	−5.42	0.002
*CD1D*	−4.54	2.05 × 10^−05^	*PDK4*	−5.28	7.81 × 10^−11^	*LOC387810*	−5.41	0.0001
*NLRC4*	−4.44	1.59 × 10^−05^	*CLEC7A*	−5.24	5.99 × 10^−12^	*UMODL1-AS1*	−5.32	3.56 × 10^−05^

**Table 5 nutrients-13-03116-t005:** The effect of dietary supplementation with the test oils on mitogen-induced changes in the expression of the top 25 most altered pathways.

Pathway	−log(*p*)	Ratio	No Overlap with Dataset (Number, %)
Baseline			
Kinetochore Metaphase Signalling Pathway	14.4	0.43	51/101 (50%)
Hepatic Fibrosis/Hepatic Stellate Cell Activation	11.5	0.31	115/186 (62%)
Cardiac Hypertrophy Signalling (Enhanced)	11.1	0.22	343/497 (69%)
Th1 and Th2 Activation Pathway	11	0.31	103/171 (60%)
STAT3 Pathway	10.6	0.33	82/135 (61%)
HMGB1 Signalling	10.0	0.30	105/165 (64%)
IL-10 Signalling	9.6	0.41	37/70 (53%)
Role of BRCA1 in DNA Damage Response	9.4	0.39	46/80 (57%)
Hepatic Cholestasis	8.5	0.27	122/186 (66%)
IL-17 Signalling	8.4	0.27	123/187 (66%)
Airway Pathology in Chronic Obstructive Pulmonary Disease	8.0	0.31	66/118 (56%)
GADD45 Signalling	8.0	0.68	5/19 (26%)
Molecular Mechanisms of Cancer	7.9	0.22	280/400 (70%)
Breast Cancer Regulation by Stathmin1	7.8	0.20	431/590 (73%)
Cell Cycle Control of Chromosomal Replication	7.7	0.41	30/56 (54%)
Aryl Hydrocarbon Receptor Signalling	7.6	0.29	88/143 (62%)
Tumor Microenvironment Pathway	7.5	0.27	113/176 (64%)
Hereditary Breast Cancer Signalling	7.4	0.29	93/140 (66%)
Role of Cytokines in Mediating Communication between Immune Cells	7.3	0.41	27/54 (50%)
Erythropoietin Signalling Pathway	7.3	0.27	114/173 (66%)
Role of Macrophages, Fibroblasts and Endothelial Cells in Rheumatoid Arthritis	7.2	0.22	220/314 (70%)
PI3K/AKT Signalling	6.9	0.26	129/184 (70%)
Th2 Pathway	6.7	0.28	87/136 (64%)
Cell Cycle: G2/M DNA Damage Checkpoint Regulation	6.7	0.41	26/49 (53%)
Oestrogen-Mediated S-phase Entry	6.7	0.54	11/26 (42%)
Post-FO supplementation			
Kinetochore Metaphase Signalling Pathway	14.6	0.42	6/101 (6%)
STAT3 Pathway	12.1	0.34	15/135 (11%)
Th1 and Th2 Activation Pathway	10.3	0.29	21/171 (12%)
IL-10 Signalling	10.2	0.41	9/70 (13%)
Cardiac Hypertrophy Signalling (Enhanced)	9.8	0.21	103/497 (21%)
Hepatic Fibrosis/Hepatic Stellate Cell Activation	9.4	0.27	60/186 (32%)
Role of Pattern Recognition Receptors in Recognition of Bacteria and Viruses	9.3	0.29	35/154 (23%)
HMGB1 Signalling	9.3	0.29	27/165 (16%)
Altered T Cell and B Cell Signalling in Rheumatoid Arthritis	9.2	0.36	19/90 (21%)
IL-17 Signalling	8.8	0.27	60/187 (32%)
Airway Pathology in Chronic Obstructive Pulmonary Disease	8.7	0.31	46/118 (39%)
Hepatic Cholestasis	8.4	0.26	50/186 (27%)
Cell Cycle Control of Chromosomal Replication	8.2	0.41	0/56 (0%)
Oestrogen-Mediated S-phase Entry	8.0	0.58	0/26 (0%)
Role of BRCA1 in DNA Damage Response	7.9	0.35	2/80 (3%)
Role of Macrophages, Fibroblasts and Endothelial Cells in Rheumatoid Arthritis	7.9	0.22	62/314 (20%)
Aryl Hydrocarbon Receptor Signalling	7.8	0.28	26/143 (18%)
Erythropoietin Signalling Pathway	7.6	0.26	28/173 (16%)
Diff. Reg. of Cytokine Production in Macrophages and T Helper Cells by IL-17A and IL-17F	7.5	0.67	1/18 (6%)
Molecular Mechanisms of Cancer	7.4	0.20	45/400 (11%)
Crosstalk between Dendritic Cells and Natural Killer Cells	7.4	0.33	11/89 (12%)
TREM1 Signalling	7.3	0.35	13/75 (17%)
GADD45 Signalling	7.1	0.63	0/19 (0%)
Role of Cytokines in Mediating Communication between Immune Cells	7.0	0.39	25/54 (46%)
Protein Ubiquitination Pathway	6.9	0.22	27/273 (10%)
Post t*CS*O supplementation			
Kinetochore Metaphase Signalling Pathway	14.4	0.43	6/101 (6%)
IL-10 Signalling	11.1	0.44	9/70 (13%)
Th1 and Th2 Activation Pathway	11	0.31	19/171 (11%)
Hepatic Fibrosis/Hepatic Stellate Cell Activation	10.5	0.30	50/186 (27%)
Airway Pathology in Chronic Obstructive Pulmonary Disease	10.4	0.35	42/118 (36%)
Hepatic Cholestasis	9.9	0.29	46/186 (25%)
Diff. Reg. of Cytokine Production in Macrophages and T Helper Cells by IL-17A and IL-17F	9.8	0.78	1/18 (6%)
Role of Pattern Recognition Receptors in Recognition of Bacteria and Viruses	9.6	0.31	31/154 (20%)
STAT3 Pathway	9.4	0.32	11/135 (8%)
IL-17 Signalling	8.9	0.28	55/187 (29%)
Differential Regulation of Cytokine Production in Intestinal Epithelial Cells by IL-17A and IL-17F	8.7	0.65	4/23 (17%)
Granulocyte Adhesion and Diapedesis	8.2	0.28	67/173 (39%)
Role of Cytokines in Mediating Communication between Immune Cells	8.0	0.43	22/54 (41%)
Role of BRCA1 in DNA Damage Response	8.0	0.36	1/80 (1%)
HMGB1 Signalling	8.0	0.28	24/165 (15%)
Altered T Cell and B Cell Signalling in Rheumatoid Arthritis	7.9	0.34	20/90 (22%)
Airway Inflammation in Asthma	7.9	0.53	3/32 (9%)
Cardiac Hypertrophy Signalling (Enhanced)	7.8	0.20	92/497 (19%)
Systemic Lupus Erythematosus in B Cell Signalling Pathway	7.8	0.24	43/275 (16%)
Th2 Pathway	7.7	0.29	15/136 (11%)
Aryl Hydrocarbon Receptor Signalling	7.6	0.29	23/143 (16%)
Cell Cycle: G2/M DNA Damage Checkpoint Regulation	7.5	0.43	0/49 (0%)
PI3K/AKT Signalling	7.3	0.26	12/184 (7%)
Erythropoietin Signalling Pathway	7.3	0.27	23/173 (13%)
Role of Hypercytokinemia/Hyperchemokinemia in the Pathogenesis of Influenza	7.2	0.34	27/86 (31%)

## Data Availability

The datasets generated during the current study are available from the corresponding author on reasonable request.

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
