# Peer review of "Dietary Supplementation with Transgenic Camelina sativa Oil Containing 20:5n-3 and 22:6n-3 or Fish Oil Induces Differential Changes in the Transcriptome of CD3+ T Lymphocytes"

_nutrients, 2021, doi:10.3390/nu13093116_

Round 1

Reviewer 1 Report

Dietary supplementation with transgenic Camelina sativa oil containing 20:5n-3 and 22:6n-3 or fish oil induces differential changes in the transcriptome of CD3+ T lymphocytes.

The authors investigated effect of supplementation of transgenic Camelina sativa (tCSO) seed oil and fish oil (FO) on composition of fatty acids of PBMCs and on the CD3+ T cell transcriptome.

  1. I still believe that the metabolism between the ages of 18 and 75 is completely different, and may translate into the results of the transcriptome study.
  2. Table 3 – The content of EPA and DHA was relatively high, therefore, why their change in composition in PBMCs are so tiny? Can authors relate to this?
  3. The effect of supplementation of transgenic Camelina sativa seed oil and fish oil (FO) was studied, which differs composition of fatty acids. In table 4 authors need to determined which gens are associated with lipogenic enzymes and enzymes of oxylipins (cyclo –, lipoxygenases, CYP 450).
  4. In supplementary material was added new table. Indeed, there was not observed any significantly effect of dietary supplementation with tCSO or FO on the fatty acid composition of PBMCs.
  5. Supplementary Figure 1. Authors described 5 genes, in which significant difference between stimulated and unstimulated cells in the same direction by both techniques were detected. Can authors give any conclusions combining these results with the administration of fatty acids?

Reviewer 2 Report

The manuscript describes the effect seed oils of a flowering plant has on the differential regulation of certain t-lymphocyte transcriptome. Although the paper in this regard the paper almost made a convincing case, there are some minor concern that need to be addressed

Minor corrections:

  1. Page 5, line 135 - give the name of the company from where RNAlater was purchases
  2.  the explanation for seeing the most downregulated genes is not very clear. It has been mentioned that this is due to 'anomalies' due to contamination by particular cells. Because the justification is being applied to two most downregulated genes, there should be sufficient  evidence provided otherwise to substantiate this claim. The author suggested that reduction by the contaminating cell population may solve this problem (page 16, line 396 and 397). It would be wise to prove this claim at least for one of the most downregulated genes to make it a convincing explanation. 
  3. Figure 3: It is a convincing figure is o show that the effect of supplementation by either FO or t CSO is almost the same to a convincing level. However looking at the figure it details it turns out that this is the case in case of general 'molecular mechanism of cancer pathways' but strikingly opposite in 'breast cancer regulation' and 'Stat3 pathway'. However there is no explanation for this discrepancy in the discussion. Therefore, a couple of sentences describing the reason behind it should be included in the discussion. 

Other than the above three comments, the manuscript seemed overall well written and provided that the above concerns are discussed well in the revision - 

Round 2

Reviewer 1 Report

The explanations of the authors are correct. 

This manuscript is a resubmission of an earlier submission. The following is a list of the peer review reports and author responses from that submission.

Round 1

Reviewer 1 Report

1. Please change qRTPCR into RT-qPCR
2. Validation of IL-17F, IL-22, IL-17A and IL-5 by either intracellular staining of in vitro stimulated cells or by measurment in supernatant
3. Inflammation/infection during the study - were participants anyhow tested for any infection during the study? This could signfiicantly afftect the overall results
4. DATA MISSING: The percentage of main T cell populations should be tested with Flow Cytometry to compare before-after and betwwen FO and tCSO. I suggest testing the % of CD4+, CD8+, gamma-delta T, MAIT and CD56+/CD3+ NKT-like cells. IF possible also iNKT cells would be of high value.
5. DATA MISSING: Moreover, please provide data indicating the post-MACS purity of cells. CD163 is listed among top-downregulated genes. CD163 is expressed on myeloid cells. Thus, this implicates impurity of MACS-isolated cells. Similary, CD1d is listed even higher - this is a protein typical for APC cells. 
6. "altered the capac-ity of T cellsto undergo changes in the expression of the transcriptome in a manner that suggests  altered  regulation  of  the  immune  response." - what does it mean?
7. Conclusions have to be totally rewritten - please address the hypothesis. The main axis fo the study was comparison of two different sources of 20:5n-3 and 22:6n-3. This is a valid question, especially when one considers lack of sustainability of modern fishing industry. Thus, please clearly address hypothesis in discussion and conclusions. 
8. Please clearly state limitations of the current study including low number of participants, lack of functional studies, bulk sequencing.

Reviewer 2 Report

Dietary supplementation with transgenic Camelina sativa oil containing 20:5n-3 and 22:6n-3 or fish oil induces differential changes in the transcriptome of CD3+ T lymphocytes.

The authors investigated effect of supplementation of transgenic Camelina sativa (tCSO) seed oil and fish oil (FO) on composition of fatty acids of PBMCs and on the CD3+ T cell transcriptome.

This work is very interesting, timely and vel planned. Also the results are very good describes and well discussed, moreover the limitations of study were described, however several points must we explained.

  1. Insufficient research group was subjected to the supplementation with tCSO and FO, moreover, why such a wide range of age was taken for this study (18-75 years)? The value of SEM is large and I suppose that a large scatter in the  results was recorded. Lipid metabolism (and not only) is very different in 18 and 75 years old people, this may translate into the results of the transcriptome study.
  2. Table 2 – Acids 18:2n-6 and 18:3n-3 are essential fatty acids (EFA), because they cannot be synthetized by human organism. However, they are substrate for the production of the other PUFA n-3 and n-6. Among polyunsaturated fatty acids n-6 exist many proinflammatory acids, which are produced proinflammatory eicosanoids (for example arachidonic acid, 20:4n-6). The subject of n-6 was not discussed in the discussion. What is more, if in tCSO was several times more 18:2n-6 than in fish oil, is it planned to investigate the long-term effect of tCSO oil supplementation?
  3. Table 3 – The content of EPA and DHA was relatively high, therefore, why their change in composition in PBMCs are so tiny?
  4. The effect of supplementation of transgenic Camelina sativa seed oil and fish oil (FO) was studied, which differs composition of fatty acids. In table 4 authors need to determined which gens are associated with lipogenic enzymes and enzymes of oxylipins (cyclo –, lipoxygenases, CYP 450).

Round 2

Reviewer 1 Report

Based on the manuscript and authors' response to my previous comments, I can see some serious methodological problems. Authors are not willing to either address them by proper testing or even acknowledge their existence . Those problems are:
1. There was no REAL purity check after MACS-separation. Although MACS-separation usually yields a decent purity (>90%), this is not always the case - purities around 70-80% are not impossible. Thus, proper testing, at least with ONE (!) colour flow cytometry with anti-CD3 staining is a PREREQUISITE for further procedures... Assessing the purity on the basis of FSC/SSC gives little to no information about real purity.
2. Second biggest problem is the stability of T cell compartment over time. I do not expect any important difference from supplementation, but over the course of the study, some significant changes in T cell subsets can occur. 14 days after infection, the composition of T cell compartment may still vary significantly from baseline e.g. in terms of gamma-delta T cells.
3. I would still insist on additional experiments with different methods at least for cytokines like IL-17 and IL-5 to see if the differential expression in RNAseq holds any real importance.

What I meant in point 6 was that this sentence should be rewritten as it hardly had any real meaning. 

Author Response

Open Review

English language and style

( ) Extensive editing of English language and style required
( ) Moderate English changes required
(x) English language and style are fine/minor spell check required

Response:  We have checked again and corrected any remaining typographical errors

( ) I don't feel qualified to judge about the English language and style

Yes

Can be improved

Must be improved

Not applicable

Does the introduction provide sufficient background and include all relevant references?

(x)

( )

( )

( )

Is the research design appropriate?

( )

( )

(x)

( )

Are the methods adequately described?

( )

(x)

( )

( )

Are the results clearly presented?

( )

(x)

( )

( )

Are the conclusions supported by the results?

( )

(x)

( )

( )

Comments and Suggestions for Authors

Based on the manuscript and authors' response to my previous comments, I can see some serious methodological problems. Authors are not willing to either address them by proper testing or even acknowledge their existence . Those problems are:
1. There was no REAL purity check after MACS-separation. Although MACS-separation usually yields a decent purity (>90%), this is not always the case - purities around 70-80% are not impossible. Thus, proper testing, at least with ONE (!) colour flow cytometry with anti-CD3 staining is a PREREQUISITE for further procedures... Assessing the purity on the basis of FSC/SSC gives little to no information about real purity.

We are pleased to be able to reassure the reviewer that we have assessed the proportion of CD3+ cells in the T cell preparation, which was in the range 80.4% to 87.7% .  We were not able to provide this information previously because the researcher who carried out the analysis is on maternity leave and cannot be contacted.  However, we now have access to the record of these measurements and have added this information in lines 129 to 130 of the revised manuscript. 

  1. Second biggest problem is the stability of T cell compartment over time. I do not expect any important difference from supplementation, but over the course of the study, some significant changes in T cell subsets can occur. 14 days after infection, the composition of T cell compartment may still vary significantly from baseline e.g. in terms of gamma-delta T cells.

The reviewer suggests what could happen, but has not provided any evidence to show that this is likely to be a primary cause of the effects on the transcriptome reported in this study.  We repeat that, as far as can be ascertained by our Clinical Research Facility which exists within a health service setting and is MHRA compliant, participants were not ill in the period prior to or during the supplementation. Thus it is reasonable to conclude that changes in the transcriptome that we report are due to the food oils used.  In this context, the outcomes of previous studies of the effects of food oil supplementation on the expression of the transcriptome or candidate genes that are cited in the manuscript support the view that important differences [in the transcriptome] may be expected. 

There were no cases of serious or prolonged infection during the study and it is unclear what actions the reviewer would have liked us to have undertaken to have absolute certainty that the participants were not experiencing an infection at the time of blood sampling which could have accounted for the effects on the transcriptome that were observed.

  1. I would still insist on additional experiments with different methods at least for cytokines like IL-17 and IL-5 to see if the differential expression in RNAseq holds any real importance.

We would like to politely remind the reviewer that the aim of the study was to investigate effects on the transcriptome, not on the downstream production of cytokines, which has been previously reported by ourselves and others after omega-3 supplementation.  Because the expression of the transcriptome changes as part of the T cell activation process, we argue that investigating the effect of dietary supplementation on the transcriptome per se is of scientific value.  The project is finished, the researcher is on maternity leave, there are no samples in which we could measure cytokine concentrations, no funds to support such analysis and the timeframe for submitting a revised manuscript (4 days) is too short to allow additional experimentation.  For these reasons we are unable to carry out the additional work requested by the reviewer while maintaining that the outcomes we have reported are still valid in addressing the aim of the study.  We have already included the lack of functional assessment as a potential limitation of the study in the previous revision of the manuscript (L413-415).    

  1. What I meant in point 6 was that this sentence should be rewritten as it hardly had any real meaning. 

Response:  We have rewritten this sentence to read “…  modified the activation-induced changes in the expression of the T cell transcriptome and so may alter the regulation of their function.”.   Lines 446 – 448. 
